# Modulators of Wnt Signaling Pathway Implied in Dentin Pulp Complex Engineering: A Literature Review

**DOI:** 10.3390/ijms231810582

**Published:** 2022-09-13

**Authors:** Marion Florimond, Sandra Minic, Paul Sharpe, Catherine Chaussain, Emmanuelle Renard, Tchilalo Boukpessi

**Affiliations:** 1Laboratory of Orofacial Pathologies, Imaging and Biotherapies, School of Dentistry, Laboratoire d’Excellence INFLAMEX, Université Paris Cité, URP 2496, 1 Rue Maurice Arnoux, 92120 Montrouge, France; 2Dental Department, Charles Foix Hospital, AP-HP, 94200 Ivry sur Seine, France; 3Centre for Craniofacial and Regenerative Biology, Faculty of Dentistry, Oral & Craniofacial Sciences, King’s College London, London SE1 9RT, UK; 4Dental Department, and Reference Center for Rare Diseases of Calcium and Phosphorus Metabolism, Bretonneau Hospital, AP-HP, 75018 Paris, France; 5Inserm, UMR 1229, RMeS, Regenerative Medicine and Skeleton, Nantes Université, ONIRIS, 44000 Nantes, France; 6CHU de Nantes, Service d’Odontologie Restauratrice et Chirurgicale, 44000 Nantes, France; 7Dental Department, Pitié Salpétrière Hospital, DMU CHIR, AP-HP, 75013 Paris, France

**Keywords:** Wnt signal, dentin pulp complex regeneration engineering, small molecules

## Abstract

The main goal of vital pulp therapy (VPT) is to preserve the vitality of the pulp tissue, even when it is exposed due to bacterial invasion, iatrogenic mechanical preparation, or trauma. The type of new dentin formed as a result of VPT can differ in its cellular origin, its microstructure, and its barrier function. It is generally agreed that the new dentin produced by odontoblasts (reactionary dentin) has a tubular structure, while the dentin produced by pulp cells (reparative dentin) does not or has less. Thus, even VPT aims to maintain the vitality of the pulp. It does not regenerate the dentin pulp complex integrity. Therefore, many studies have sought to identify new therapeutic strategies to successfully regenerate the dentin pulp complex. Among them is a Wnt protein-based strategy based on the fact that Wnt proteins seem to be powerful stem cell factors that allow control of the self-renewal and proliferation of multiple adult stem cell populations, suitable for homeostasis maintenance, tissue healing, and regeneration promotion. Thus, this review outlines the different agents targeting the Wnt signaling that could be applied in a tooth environment, and could be a potential therapy for dentin pulp complex and bone regeneration.

## 1. Introduction

The main goal of vital pulp therapy (VPT) is to preserve the vitality of the pulp tissue, even when it is exposed due to bacterial invasion, iatrogenic mechanical preparation, or trauma [1]. VPT procedures consist of direct pulp capping, and partial or full pulpotomy with bioactive capping materials. Calcium hydroxide (CH) has been extensively used for direct pulp capping and has long been considered the “gold standard” [2]. CH can release hydroxyl and calcium ions that create an alkaline bactericidal environment around the pulp tissues, prompting the formation of necrotic tissue beneath the exposed pulp, and this tissular reaction may lead to increases in cell differentiation, collagen secretion, and dentin formation [3,4].

However, the poor quality of the resulting dentin bridge and its lack of sealing within the dentin walls explain why some authors prefer the use of Tricalcium silicate-based (TCS-based) cements [4,5], such as ProRoot White MTA (Dentsply, Tulsa Dental, Tulsa OK, USA) or Biodentine (Septodont, Saint-Maur-des-Fossés, France), which both have high clinical success rates in dentistry. The widespread clinical indications of TCS-based cements are mainly based on their ability to form CH as a by-product of hydration. MTA has been considered as potential gold standard for vital pulp therapy, because it integrated better with the pulp tissues than CH [5] and showed greater success than other capping agents when used in different conditions in clinical trials [6,7]. The type of new dentin formed can differ in different manners (cellular composition and function). Even though there is no consensus regarding the origin of this new tissue [7], it is generally admitted that reactionary dentin produced by odontoblasts has a tubular structure, while the reparative dentin does not or has much less [8].

VPT aims to maintain and, if possible, regenerate the vitality of dental pulp. This last objective is not achieved using the conventional VPT procedures. Therefore, many studies have sought to identify new therapeutic strategies to successfully regenerate the dentin pulp complex. Two different ways are currently being tested to achieve this enormous objective. The first involves the introduction of stem or progenitor cells into a site of damage, and the second aims to activate endogenous stem cells to promote tissue regeneration. This last strategy, in which endogenous stem cells are activated, avoids the risk associated with the implementation of different cells in nature or reprogrammed cells into the human body. For this purpose, a Wnt protein-based strategy, based on the fact that Wnt proteins are powerful stem cell factors [9], allows control of the self-renewal and proliferation of multiple adult stem cell populations [10].

The secreted family Wnt proteins participate in the regulation of cell differentiation, proliferation, and apoptosis, and through these mechanisms, play a key role in tissue generation, regeneration, and self-renewal [11]. Wnts induce intracellular signaling by binding to the extracellular domain of receptors encoded by Frizzled (Fz). These proteins interact also with the low-density lipoprotein receptor-related protein (LRP) 5 and LRP 6 transmembrane proteins that act as coreceptors for Wnts. They also linked to neurotrophic tyrosine kinase, receptor-related 2 (NTRK2). By means of an intracellular signal transduction pathway (activation of Dishevelled (DVL)), a protein of the destruction complex prevents activation of the destruction complex, constituted of Axin, adenomatosis polyposis coli (APC), glycogen synthase kinase 3 (GSK3), and other factors. At the same time, the cytoplasmic domain of LRP5/6 becomes phosphorylated and binds axin. This leads to disassembly of the APC-axin- β catenin complex and the release of β- catenin. Then, β-catenin can accumulate in the cytoplasm and eventually, β-catenin translocates to the nucleus, where it acts as a transcriptional activator of transcription factors in the T-cell-specific factor/lymphoid enhancing factor Tcf/Lef family and increases the transcription of Wnt target genes encoding axin, Smad6, cyclin D1 and Cx43 [12] (Figure 1). There are numerous Wnt ligands, receptors, co-factors, antagonists, and intracellular mediators.

As the regenerative capacity of multiple mammalian tissues has been shown depend-ing on Wnt/β- catenin signaling and its activation. In the context of pulp dentin, complex regeneration the role of canonical pathway seems obvious. Numerous studies have shown this role in reactionary dentinogenesis [13]. In reparative dentinogenesis, the repair process is accompanied by increased Axin2 expression, which results in differentiation of Axin2 expressing cells from resident dental pulp stem cells into odontoblasts-like cells [14].

However, the specific role of Wnt/β-catenin signaling on odontoblast-like differentiation of hDPSCs is not completely known. Indeed, some authors such as Scheller et al. demonstrated that canonical Wnt signaling inhibited odontogenic differentiation of hDPSCs [15]. In addition, Zhang et al. reported that Wnt10a, a Wnt agonist, could negatively regulate the differentiation of DPSCs into odontoblasts by down-regulating odontoblast specific genes [16]. However, some other researchers have reported that β-catenin accumulation by various agonists promoted odontoblastic differentiation in hDPSCs [17,18,19,20]. Thus, β-catenin could play an essential role in tertiary dentinogenesis [17,21]. β-catenin could act as an activator of the transcription factor runt-related transcription factor 2 (Runx2) to enhance the odontoblastic differentiation of dental pulp stem cells [19] and stem cells from the apical papilla (SCAP) [22].

In this review, we aimed to recapitulate new biologically based strategies to enhance this natural repair response by regulating Wnt signaling via modulatory molecules. These molecules could be a potential therapy target for dentin pulp complex.

## 2. Modulators of Wnt Beta-Catenin Signaling Acting on Dental Pulp Cells

### 2.1. Inorganic Calcium-Containing Materials

The treatment of dental caries that results in pulp exposure is currently managed by replacing lost dentine with inorganic calcium-containing materials such as CH, MTA or Biodentine that remain in the crown. Since this dentine is formed directly from new odontoblast-like cells that differentiate from resident stem cells in the pulp [14], it is imaginable that overstimulation of stem cell activity might result in increased odontoblast differentiation resulting in more efficient regenerative dentine formation. The most studied material (ProRoot MTA), in direct contact with DPSCs/DPCs, has shown significant positive results in in vitro assays assessing the involvement of the MAPK subfamilies JNK and P38, the ERK subfamily, the nuclear factor kappa B (NF-κB), and Wnt/β-catenin pathways.

### 2.2. Small Molecule GSK3 Inhibitors

In the context of response to damage, the pulp dentin complex can induce a Axin2 increase repair process in odontoblast-like cells that subsequently form reparative dentin [14]. Glycogen synthase kinase 3 (GSK3) is a core intracellular component of the Wnt/β-catenin signaling pathway that phosphorylates Axin and β-catenin [23,24,25]. When there are Wnt ligands, GSK3 activity is disabled, and β-catenin can enter the nucleus to interact with Lef/Tcf transcription and express the target genes such as Axin2. When Wnt ligands are absent, β-catenin and Axin2 are phosphorylated, causing their ubiquitination and then their degradation. GSK3 inhibitors (which are also Wnt pathway activators) could have various forms. They have shown to have natural or synthetic sources and display different mechanisms of action. A range of small molecule antagonists of GSK3 have been developed as drugs to activate the Wnt pathway in responsive cells [23,26,27,28]. Several GSK3 inhibitors have been shown to promote dentin repair in mice and rats with experimental pulp exposure [29,30].

#### 2.2.1. Tideglusib

Tideglusib is the most studied GSK3 drug that has to date been shown to be safe in patients [31]. Delivery of GSK3 inhibitor drugs (20 μM CHIR99021 and 1 μM Tideglusib on biodegradable collagen sponges) directly into experimentally exposed pulp cavities in mice results in upregulation of Wnt-activity in pulp stem cells [29] and induction of high quality of reparative dentinogenesis [30]. This reparative dentine was biochemically indistinguishable from native dentine when analyzed by Raman spectroscopy. Although the extent of damage in rats is not comparable to that in large lesions in humans, the successful scaleup of reparative dentine formation in vivo seems promising, highlighting the potential of this approach. However, Tideglusib has low aqueous solubility, and in clinical trials, is delivered in a granulate form suspended in water that is less suitable in clinical practice.

#### 2.2.2. NP928

NP928 is a new GSK3 inhibitor small-molecule drug that has increased aqueous solubility compared to other thiadiazolidinone (TDZD) drugs and can activate Wnt/β- catenin pathway similarly to tideglusib. Therefore, NP928 is a modified version of Tideglusib that removes the naphthyl moiety and increases solubility. The reparative potency and clinical usability of NP928 was evaluated in microdose concentrations of loaded MA-HA hydrogels in a pulp damage model in wild-type mice, and showed more reparative dentine was in tested groups than in controls [32].

The use of hydrogel looks superior to the sponge delivery, and the overall simpler user experience for the clinician. The use of hydrogel looks superior to the sponge delivery, and the overall simpler user experience for the clinician. These results allowed a drug called ReDent^®^ to be transferred to clinic, on the basis of it being ready for its first human clinical trials.

A tooth cavity is a good therapeutic site for the use of a small molecule, used at very tiny concentrations. They should have a short half-life and limited range of action. They do not activate cells in the roots, for example.

#### 2.2.3. Tivantinib

Tivantinib is a small molecule that inhibits c-Met receptor tyrosine kinase. It is a non-ATP competitive inhibitor. GSK3α and GSK3β, two structurally isoforms of GSK3, have been identified as new targets for this molecule [33]. These isoforms are negatively and positively regulated by serine or tyrosine phosphorylation, respectively. Recently, GSK3α and β were inhibited by Tivantinib in lung cancer cells [34]. Therefore, this molecule was tested in phase III trial of the treatment of hepatocellular carcinoma. Tivantinib’s biocompatibility and low cytotoxicity have also been demonstrated on progenitors’ murine cells [35]. It is quite interesting for the best of our knowledge and dental practice to notice that a c-Met inhibitor tyrosine kinase used in a carcinoma treatment presented a weak toxicity for dental pulp cells and the ability to activate the Wnt/β-catenin pathway at very low concentrations in vitro [35]. Nevertheless, further studies are needed to analyze and evaluate in vivo effects of delivering tivantinib on pulp injuries via collagen sponges or other vectors, such as hydrogels.

#### 2.2.4. Lithium Chloride

Lithium compounds are inhibitors of GSK3β, used among other things to treat the bipolar patients and inhibit cancer cell metastasis [36]. Many in vitro studies reported that these agents can potentiate bone regeneration process and upregulate osteoblast differentiation and mineralization [37,38]. Ishimoto’s team identified the effect of 10 mM Lithium chloride as an activator of reparative tubular dentin formation. The application of Lithium has been realized locally in rats after a pulpotomy procedure. Thus, they have shown stimulation of the Wnt/b-catenin pathway (through inhibition of the b-catenin destruction complex) when pulp cells were treated with lithium ions in vitro [39]. Interestingly, recently LiCl-100 mM was shown to activate this signaling pathway in vitro [40]. In rats, capping of pulps with surface pre-reacted glass combined with LiCl at concentrations of 10 mM or 100 mM was associated with the formation of complete reparative dentin structures that were continuous with the primary dentin without any defects, similar to that produced by MTA [40]. A research team has incorporated lithium-containing bioactive glass in a commercial GIC, so that lithium released from the GIC could naturally penetrate dentin and stimulate odontoblast activity. They succeeded to stimulate dentin formation and improve repair in a murine molar defect model [41]. Likewise, in a context injury in restorative treatment with resin polymers, the study of Bakopoulou et al. in 2015 [42] showed that human DPSCs were stimulated after lithium treatment through the accumulation of β-catenin and enhancement of its translocation in nucleus and expression of transcription factors. Exposure of lithium chloride-pre-treated cells to TEGDMA (triethylene-glycol-dimethacrylate) showed a stronger activation of the pathway. Thus, these findings stipulated that TEGDMA could continue to induce canonical Wnt pathway in DPSCs that were already “activated” by various environmental factors during pulp repair.

### 2.3. R-Spondin 2

Recombinant proteins such as the Wnt agonist R-spondin [43] have been used to treat oral mucositis [44]. R-spondins are secreted proteins that act as stem cell growth factors [45]. It has been shown that R-spondins clearly increase Wnt signal. R-spondin 2 (Rspo2) has been reported to show a predominant role in many differentiation processes, such as neurogenic differentiation [46], chondrogenic [47], and osteoblastic [48] differentiation. In a recent study, Gong Y. et al. succeeded in inducing odontogenic differentiation in combination with exogenously added Rspo2 in hDPSCs through an increase in levels of both mRNA and protein expression of dentin sialophosphoprotein, dentin matrix protein-1, alkaline phosphatase, bone sialoprotein, and protein expression levels of osteopontin and osteocalcin, whereas silencing Rspo2 significantly decreased the expression levels of these odontogenic markers [49]. This promotion of odontogenic differentiation is attributable to the activation of Wnt/β-catenin signaling. Thus, more investigations are needed to evaluate effects of Rspo2 in dental pulp complex repair.

### 2.4. Wedelolactone

Wedelolactone is a natural plant compound that has been shown to have anti-inflammatory, anticancer, and antiosteoporosis effects. The effect of wedelolactone has also been evaluated for dental treatment. For that purpose, DPSCs were treated with wedelolactone in vitro [50]. This experiment has been shown to promote odontoblast differentiation and mineralization through a direct enhancement of the nuclear accumulation of β-catenin and expression of genes involved in odontoblast differentiation. These genes included DMP-1, DSPP, and runx2. This study highlighted that wedelolactone induced the differentiation of odontoblast cells by means of semaphorin 3A/neuropilin-1 pathway-mediated β-catenin stimulation and NF-κB pathway disactivation.

### 2.5. Semaphorin 3A and Its Receptor Neuropilin 1

Sema3A has been shown to be an osteoprotective factor by inhibiting osteoclastic bone resorption and promoting osteoblastic bone formation through canonical Wnt/β-catenin signaling [51]. When Neuropilin-1 (NRP1) binds to Sema3A, it stimulates osteoblast differentiation through the classical Wnt/β-catenin pathway. Overexpression of NRP1 upregulated dentin matrix protein-1, dentin sialophosphoprotein, alkaline phosphatase protein level, and mineralization in DPSCs, while knockdown of NRP1 induced the opposite effects. NRP1, therefore, regulates DPSCs via the classical Wnt/β-catenin pathway [52]. Sema3A has also been showed to induce cell migration, chemotaxis, proliferation, and odontoblastic differentiation of DPSCs, and Sema3A application to dental pulp exposure sites in a rat model induced effective reparative dentin reconstruction [19].

### 2.6. Wnt3a Protein

In vitro, the effects of a continuous activation of Wnt/β-catenin signaling by the addition of Wnt3a on the mineralization and differentiation of pulp cells demonstrated that Wnt3a induced marked increases in the expression of Dmp1, Dspp, and Bsp, compared to controls between days 10 and 17 [53], and highlighted the role of Wnt/β-catenin signaling in the survival of resident progenitors. Indeed, a limited and early exposure to Wnt3a resulted in increased proliferation and decreased apoptosis in the undifferentiated population [54]. In vivo, the study of Hunter et al.; in 2015 [54] showed that pulp healing was positively impacted by a Wnt3a (a typical canonical Wnt ligand) amplified environment in a model of direct pulp capping. In fact, the application of Wnt3a through a lipidic vesicle, which allowed the maintenance of its activity, led to the formation of a reparative matrix resembled native dentin. In addition, this liposome delivered Wnt3a protected pulp cells from death and stimulated proliferation of undifferentiated cells in the pulp, which together significantly improved pulp healing.

### 2.7. Sclerostin

Sclerostin, is a secreted glycoprotein which is largely produced by osteocytes under a physiologic environment. It is an antagonist of the Wnt-BMPs signaling pathway through its binding to LRP 5/6receptor which is present on the membrane osteoblast [55]. It has been shown that when sclerostin is downregulated, an increase of osteogenesis and in bone mass are observed [56].

Secretion of sclerostin by odontoblasts has been demonstrated during tooth development [57,58]. Many studies have investigated the potential role of this molecule in the dentin pulp complex healing process [59]. In absence of sclerostin in Sost knock out mice, an increase in the pulp-healing process, following a direct pulp-capping mice model, was demonstrated [59]. In vitro, cultures of mDPCs isolated from Sost knock out germs allowed for elevated mineralization. Interestingly, the role of sclerostin in the process of human dental pulp cell (hDPCs) senescence was studied as the expression level of sclerostin varies in embryonic and adult mouse incisors and molars, and in aged individuals [58,60]. Thus, it has been shown that expression of sclerostin was increased in senescent human dental pulp and subculture-induced senescent hDPCs by immunohistochemistry and qRT-PCR analyses [61]. In addition, overexpression of sclerostin led to hDPCs senescence and inhibition of odontoblastic differentiation of hDPCs. Therefore, an anti-sclerostin treatment may be beneficial for the maintenance of the proliferation and odontoblastic differentiation potentials of hDPCs and to improve the pulp healing process in exposed pulps treatment. Indeed, Liao et al., 2019 have shown that sclerostin increased the inflammatory responses of odontoblasts under an LPS-induced environment and led to impaired dentin tissue regeneration by inhibition of odontoblastic differentiation of inflamed DPCs [62]. These findings allow new ideas toward therapeutic treatments combining anti-inflammatory effects and promotion of regeneration during dental pulp inflammation.

## 3. Cellular Metabolism Effect of Wnt

### 3.1. Epigenetic Remodeling in Human DPSCs under Wnt Ligant Exposure

The canonical Wnt signaling pathway is considered as an important regulator of stemness [11,23,63] and cell differentiation [15] in DPSCs and many other stem cell types [64]. The epigenetic regulations that Wnt pathways may exert on DPSCs have been studied. The authors have found that Wnt-3a exposure induced plural epigenetic reprogramming facets in DPSCs; especially, a global DNA hypomethylation, a global histone hyperacetylation, and an increase in both activating and repressing histone methylation marks were highlighted [65]. These findings could have seductive implications in the optimization of the clinical cell therapy.

### 3.2. Effect on Energetic Metabolism

Certain systemic conditions compromised the capacity of proliferation and odontogenic differentiation of human DPSCs. Diabetes is one of these pathological conditions. The reduction in human pulpal mesenchymal cells stemness in diabetic patients could affect the regenerative capacity of pulp-dentin complex and the formation of the dentin bridge. Based on these data, some authors studied the potential role of Wnt signaling in a high glucose-induced senescence model (close to diabetic conditions) [66]. Interestingly, Asghari et al. have observed the decrease in proliferation of DPSC, as well as an increased number of senescent cells and an increased p21 expression, after being exposed to different concentrations of glucose. β -catenin and Wnt1 expression in response to high glucose were significantly increased. In the same way, the authors demonstrated that in the presence of a β-catenin inhibitor PNU-74564, the amount of the senescent cells was reduced. Therefore, Wnt signaling might be the potential target for the inhibition of the senescence response in the hyperglycemic condition, suggesting the potential development of bioactive materials applied in pulp capping that would be specific for diabetic patients.

#### 3.2.1. Famotidine

Famotidine is a competitive inhibitor of the histamine receptor, which is the dominant receptor involved in gastric acid secretion. This binding prevents the activation of adenylate cyclase normally induced by histamines. Many studies have investigated the Famotidine potential anti GSK3 effect. It seems that this inhibiting role could be attributed to the hypoglycemic aspect. Indeed, H2-receptor inhibitors could affect glucose metabolism and its role in the decrease in the glycemic response curve in vivo through binging with GSK3β [67]. Its biocompatibility and low cytotoxicity have also been demonstrated on progenitors’ murine cells [35]. Its effects on DPSC could be studied in a context of glucose exposition.

#### 3.2.2. Olanzapine

Olanzapine used to be a particular pharmacological psychiatric drug used for the treatment of schizophrenia. Its potential anti GSK3 effect has been sought by many authors. As Famotidine, it seems that this inhibiting role could be due to the hypoglycemic effect. Its biocompatibility and low cytotoxicity have also been demonstrated on progenitors’ murine cells [35]. This molecule could also be studied in glucose exposition condition of DPSC.

## 4. Conclusions

The Wnt/b catenin pathway is a very complex intra cellular pathway.

Several studies have shown that Wnt signals are necessary for pulp dentin complex formation and repair, and in other studies, some research groups have succeeded in demonstrating that a Wnt stimulus is sufficient to induce tissue regeneration. Small-molecule drugs that stimulate Wnt/β-catenin have shown promise as a novel biological therapy for treating exposed pulpal lesions. It is necessary to consider the need of local application to avoid systemic side effects of the highly potent small molecules acting as Wnt agonists at very tiny concentrations. Interestingly, they are very cheap to make. Strong proof-of-concept data are still needed, however, along with well-crafted safety plans, to make regenerative dental medicine a reality.

After pulp exposure, molecules enhancing Wnt/β-catenin can be applied in contact with the pulp on different supports such as hydrogels or sponges (Figure 2). These materials must be sealed tightly with restorative material. The release of molecules stimulates the regeneration of the pulp-dentin complex.

## Figures and Tables

**Figure 1 ijms-23-10582-f001:**
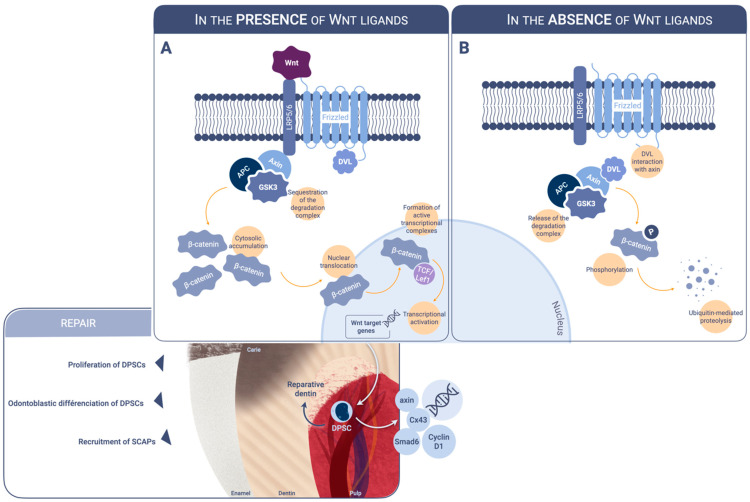
Schematic illustration of the canonical Wnt pathway. (**A**). In the presence of Wnt ligands interacting with LRP5/6 and frizzled, the β-catenin degradation complex is sequestrated. Cytosolic accumulation of β-catenin leads to nuclear translocation and binding to transcription factors in the Lef/Tcf family. The resulting active transcriptional complex controls the expression of target genes involved in tissue generation, regeneration and, self-renewal. (**B**). In the absence of Wnt ligands, the interaction between DVL and axin leads to the phosphorylation of cytosolic β-catenin by a protein complex involving APC, axin, and GSK3. β-catenin is then degraded by ubiquitin-mediated proteolysis. LRP: lipoprotein receptor-related protein. DVL: dishevelled-APC: adenomatosis polyposis coli. GSK: glycogen synthase kinase. TCF/Lef1: T-cell factor/lymphoid enhancer factor. DPSCs: dental pulp stem cells. SCAPs: stem cells from apical papilla.

**Figure 2 ijms-23-10582-f002:**
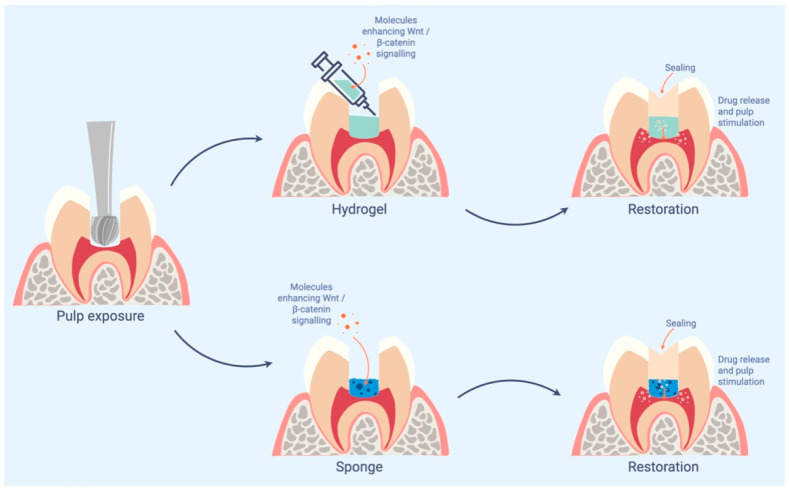
Schematic illustration of the clinical application of Wnt/β-catenin enhancers.

## Data Availability

Not applicable.

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
