# Peer review of "Modulators of Wnt Signaling Pathway Implied in Dentin Pulp Complex Engineering: A Literature Review"

_ijms, 2022, doi:10.3390/ijms231810582_

Round 1

Reviewer 1 Report

The manuscript submitted by Florimond et al. is of potential interest but needs to be modified, in my opinion, as reported below

1. Chapter 2 deals with the Wnt signal in general, but then the topic is resumed in the following paragraphs regarding what has been detected as for the stimulation of dentin regeneration. My suggestion is to include all the considerations in chapter 2, accordingly modifying the related figure.

2. The importance of treating the Wnt signal with regard to osteogenesis in dental pulp cells is unclear. I believe that chapter 3 should be better placed in the context, otherwise it is disconnected from the purpose of the review itself.

3. The authors should clarify how the Wnt signal is important for some substances that have a hypoglycemic effect.

4. Overall, it is difficult to follow the discussion regarding the importance of Wnt signal in the repair of dental pulp. There is no incisiveness in the observations produced which are mostly generic and speculative.

Minor point

on page 9, the authors forgot to delete the phrase about "Patents driving"

Reviewer 2 Report

This paper is not acceptable because of its high level of plagiarism.

Round 2

Reviewer 1 Report

I think the review has improved a lot overall. However, I have noted the following points that need further revision:

1. In Fig. 1, letter A covers a part of the figure and I don't know if that was the intention (then that part should be eliminated) or not. In the legend of the same figure, the verb "control" in line 9 should be "controls";

2. on page 6, lines 175-178, the phrase "the use of hydrogel ..." is repeated twice;

3. on page 6, lines 183-186, the two isoforms alpha and beta of GSK3 are introduced which previously had not been defined. Therefore it is necessary to briefly explain their existence;

4. in the chapter on lithium chlroride the phrase "they succeed to stimulate ..." needs to be revised, in which verbs should be used in the past tense. It is also unclear whether the effect of TEGDMA is related to new dentinogenesis or not.
